# Prolonged SARS-CoV-2 Infection and Organizing Pneumonia in a Patient with Follicular Lymphoma, Treated with Obinutuzumab—Challenging Recognition and Treatment

**DOI:** 10.3390/v15030693

**Published:** 2023-03-07

**Authors:** E. Łyżwa, M. Sobiecka, K. Lewandowska, I. Siemion-Szcześniak, I. Barańska, M. Klatt, R. Langfort, M. Szturmowicz, W. Tomkowski

**Affiliations:** 11st Department of Lung Diseases, National Research Institute of Tuberculosis and Lung Diseases, 01-138 Warsaw, Poland; 2Department of Radiology, National Research Institute of Tuberculosis and Lung Diseases, 01-138 Warsaw, Poland; 3Department of Microbiology, National Research Institute of Tuberculosis and Lung Diseases, 01-138 Warsaw, Poland; 4Department of Pathology, National Research Institute of Tuberculosis and Lung Diseases, 01-138 Warsaw, Poland

**Keywords:** prolonged SARS-CoV-2 infection, COVID-19, obinutuzumab, organizing pneumonia

## Abstract

A Severe acute respiratory syndrome coronavirus 2 (SARS-CoV-2) causing coronavirus disease 2019 (COVID-19) led to a pandemic outbreak in 2019. COVID-19’s course and its treatment in immunocompromised patients are uncertain. Furthermore, there is a possibility of protracted SARS-CoV-2 infection and the need for repeated antiviral treatment. Monoclonal antibodies against CD20, which are used, among other things, in the therapy of chronic lymphocytic leukaemia and follicular lymphoma, can induct immunosuppression. We present a case report of a patient with follicular lymphoma, treated with obinutuzumab, who was diagnosed with prolonged, ongoing SARS-CoV-2 infection and related organizing pneumonia. The recognition and the treatment were challenging which makes this case noteworthy. Antiviral therapy with several medications was administrated to our patient and their temporary, positive effect was observed. Moreover, high-dose intravenous immunoglobulin was applied, because slowly decreasing IgM and IgG levels were observed. The patient also received standard treatment of organizing pneumonia. We believe that such a complex approach can create an opportunity for recovery. Physicians should be conscious of the course and treatment possibilities facing similar cases.

## 1. Introduction

Severe acute respiratory syndrome coronavirus 2 (SARS-CoV-2) causing coronavirus disease 2019 (COVID-19) has spread throughout the whole world and led to a pandemic outbreak in 2019. Immunocompromised patients, as the most vulnerable population, have been precisely observed since the beginning of the pandemic. It is a heterogeneous group that includes both patients with humoral and cellular immune deficiencies, or a combination of both, but also patients that have received immunosuppressive treatment for different reasons, including haematological malignancies. Rituximab and obinutuzumab are monoclonal antibodies against CD20 and are used in the therapy of chronic lymphocytic leukaemia and follicular lymphoma, among others. They can induce direct cellular death and participate in antibody-dependent cellular cytotoxicity and antibody-dependent cellular phagocytosis [1].

The course and treatment of COVID-19 in immunocompromised patients are uncertain. Furthermore, there is a possibility of protracted SARS-CoV-2 infection and the need for repeated antiviral treatment. We present a case report of a patient with follicular lymphoma, treated with obinutuzumab, who was diagnosed with prolonged, ongoing SARS-CoV-2 infection. He has never been treated in an intensive care unit (ICU); however, he required repeated, long hospitalisations and because of temporary revealing respiratory failure, also required oxygen therapy. Both the diagnostics and treatment were challenging and demanded a literature review. Off-label treatment was consequently applied.

## 2. Case Report

A 51-year-old, non-smoking, male with a history of Ehlers-Danlos syndrome and follicular lymphoma was admitted to our department in May 2022 due to a one-month-long persisting fever of up to 40 degrees Celsius, productive cough, and dyspnoea. The lymphoma was treated with chemotherapy including cyclophosphamide, doxorubicin, vincristine, and prednisone combined with the monoclonal antibody obinutuzumab. He received six courses of such immunochemotherapy; the last course was in May 2021. He then received maintenance therapy with obinutuzumab in monotherapy; the last dose was in March 2022. According to the positron emission tomography-computed tomography (PET-CT) scan, remission was achieved. The patient provided a history of having self-tested positive for SARS-CoV-2 at home using a rapid antigen test in April 2022. He did not require medical care because the symptoms were mild. In May his condition deteriorated, and the symptoms mentioned above appeared. Some empiric antibiotic therapy was administrated using amoxicillin with clavulanic acid, levofloxacin, clarithromycin, and doxycycline, but no improvement was observed. Upon haematological consultation, lymphoma relapse was thought to be highly unlikely, and PET-CT was again recommended. Pending examination, on the 50th day after the SARS-CoV-2 antigen test was performed and appeared positive, the patient reported to our hospital. On admission to our department, the patient was in average condition. Blood pressure, heart rate, and oxygen saturation were within normal ranges. Lung auscultation revealed fine crackles bilaterally. Laboratory analysis showed elevated C-reactive protein (CRP) at 48.9 mg/L (normal: <5 mg/L) and ferritin level of 2512.46 mg/mL (normal: <274.66 mg/mL); decreased IgM level of 25 mg/dL (normal: 40–230 mg/dL); neutropenia, white blood cells 3.1 × 10^9^/L (normal: 4.23–9.07 × 10^9^/L) with lymphopenia, 0.52 × 10^9^/L (normal: 1.32–3.57 × 10^9^/L); and mild anaemia, haemoglobin 12 g/dL (normal:13.7–17.5 g/dL); other blood tests were normal. Sputum, blood, and urine cultures were negative (Table 1).

Antinuclear antibodies (ANA) were undetectable. A nasopharyngeal swab (NPS) was negative for SARS-CoV-2 by reverse-transcription polymerase chain reaction (RT-PCR). RT-PCR tests were analysed at the Department of Microbiology in our hospital with GeneXpert IV (Xpert^M^ Xpress SARS-CoV-2, Cepheid). Moreover, the patient provided a history of a 3-dose vaccination against SARS-CoV-2. Computed tomography (CT) scans performed before admission were overviewed by experienced radiologists and organizing pneumonia due to obinutuzumab treatment was suspected, lesions were not characteristic of the COVID-19 pattern. Moderate restriction with moderately decreased transfer factor for carbon monoxide (TLco) was observed in pulmonary function tests (PFTs) (Table 2).

Bronchoscopy with cryobiopsy was performed, and material for microbiological tests and histopathological evaluation was collected. The culture was sterile. The smear of bronchoalveolar lavage fluid was negative for acid-fast bacteria (final cultures were also negative), and there was no fungal growth. Histopathological examination showed lesions typical for organizing pneumonia (Figure 1 and Figure 2).

After the bronchoscopy, the patient’s condition deteriorated, and respiratory failure requiring oxygen therapy developed. Moreover, haemoptysis was observed. Contrast-enhanced CT with pulmonary angiogram excluded pulmonary embolism and showed multiple bilateral and mostly peripheral areas of consolidation and ground-glass opacities suggesting an organizing pneumonia pattern (Figure 3).

Based on the histopathological analysis and corresponding clinical and radiological picture, the diagnosis of organizing pneumonia due to obinutuzumab treatment was established. Methylprednisolone was administered in intravenous pulses of 250 mg per dose followed by 0.75 mg per kg of prednisone administrated orally. Intravenous antibiotic therapy was also applied, taking elevating inflammatory markers into consideration. As a result of the treatment the patient’s condition improved, the features of respiratory failure subsided, and he was discharged with a recommendation for prednisone treatment continuation with gradual dose tapering.

On day 94, from the initial positive SARS-CoV-2 test, the patient presented to our department again, due to a fever relapse (febrile episodes several times a day, up to 40 degrees Celsius). Meanwhile, PET-CT was performed, and proved lymphoma remission. On admission to our department, his condition was good, oxygen saturation was 97% on room air. On auscultation, crackles were found, and laboratory tests showed inflammatory parameters were slightly elevated. NPS antigen and RT-PCR tests were negative for SARS-CoV-2 (SARS-CoV-2 antigen BioMaxima Ag Rapid Tests were used in our hospital). CT scans showed both partial regression of some lesions and progression of others with new consolidations that confirmed organizing pneumonia (Figure 4).

Methylprednisolone in intravenous pulses of 250 mg per dose and then higher prednisone doses, as at the beginning, were again administrated. After an initial improvement, the fever relapsed. An NPS antigen test revealed a positive result for SARS-CoV-2. A five day-long therapy with remdesivir (the initial one-day dose of 200 mg, followed by four-days of therapy with 100 mg) was then performed. Consequently, the patient’s condition improved, and no laboratory abnormalities apart from elevated ferritin level were found. He was discharged afebrile with a recommendation, as before, for steroid therapy continuation.

On day 123 the patient was readmitted to our department again because of fever relapse and dyspnoea. The NPS antigen test, performed on admission, was positive for SARS-CoV-2. Laboratory tests revealed lymphocytopenia, mildly elevated CRP, D-dimers, and as high a ferritin level as before. Contrast-enhanced CT with pulmonary angiogram excluded pulmonary embolism and showed new peribronchial ground-glass lesions. Differential diagnostics with wide microbiological analysis were again performed and no other microbiological agents were found; despite that, empirical, wide spectrum, intravenous antibiotic therapy was administrated. Moreover, after the NPS antigen SARS-CoV-2 test became negative, methylprednisolone pulses of 250 mg per dose and then a high-dose prednisone course were initiated for the third time. Despite this, in a few days, a fever relapse was observed. An NPS antigen test for SARS-CoV-2 was repeated and revealed a positive result again. Relapsing SARS-CoV-2 infection resulting in organizing pneumonia was finally diagnosed. Antibiotic therapy was withdrawn, and prednisone doses were gradually reduced. One-day therapy with 600 mg casirivimab and 600 mg imdevimab, and ten-day therapy with concomitant remdesivir (the initial one-day dose of 200 mg, followed by a nine-day therapy with 100 mg) were administrated. Afterwards, he was discharged in a good condition, and was afebrile.

On day 179 the patient was readmitted to our department as scheduled to perform a check-up and to administrate the next doses of casirivimab and imdevimab, as a prophylaxis in the immunocompromised patient. Interestingly, after the antiviral treatment administrated during the recent hospitalisation, the ferritin level was within normal ranges. CT scans showed an unambiguous partial regression of recently described ground-glass opacities. On admission day, fever episodes relapsed. The NPS results were negative for SARS-CoV-2 based on antigen and RT-PCR tests. After other infectious agent exclusion, a one-day therapy with 300 mg of casirivimab and 300 mg of imdevimab was applied as scheduled. Despite that, the improvement was not achieved. Bronchoscopy was performed and the bronchoalveolar lavage fluid (BALF) revealed a positive result of RT-PCR for SARS-CoV-2. A ten-day therapy with remdesivir (the initial one-day dose of 200 mg, followed by nine-day therapy with 100 mg) was then administrated for the third time in our patient. Moreover, during this hospitalisation, lower than before IgM—20 mg/dL (normal: 40–230 mg/dL) and IgG—541 mg/dL (normal: 700–1600 mg/dL) levels were observed. A high-dose intravenous immunoglobulin course was administrated (0.4 g per kg). Consequently, the frequency of fever episodes decreased. After another seven days, the patient was in good condition and fever free, and was therefore discharged. Summarised clinical data of all hospitalisations are presented in Table 3.

## 3. Discussion

Since the beginning of the pandemic, some cases of COVID-19 in patients treated with anti-CD20 monoclonal antibodies (antyCD20mAB) due to haematological illnesses, have been described [3,4,5,6,7,8,9,10,11,12,13,14]. The majority of those patients received rituximab. The time from the last dose of medication and the infection with SARS-CoV-2 ranged between 2 weeks and 12 months, but in one case it was more than 14 years [6]. In some patients, the course of COVID-19 was mild at the beginning and they required only observation but after a couple of weeks, they presented to the hospital with symptoms such as elevated body temperature, dyspnoea, and dry cough [3,7,8,11,12,13]. Some patients developed respiratory failure during the disease course [3,4,5,6,7,8,10,11,12,14]. Only a few died because of severe COVID-19 courses [6,9,10,11,12,14]. The maximum duration of COVID-19 symptoms was about one year [9,15]. Our patient received obinutuzumab. Treatment with this medication can lead to even worse COVID-19 outcomes than in case of rituximab [16]. The initial COVID-19 presentation in our patient was mild. He required medical care only because the cause of later-developing symptoms was not clear. Our manuscript draws attention to the possibly relapsing and protracted course of the illness, and the need to take prolonged COVID-19 into consideration in differential diagnostics.

In our patient, CT scans showed an organizing pneumonia pattern. Radiological lesions were not characteristic of COVID-19 itself. Organizing pneumonia can be caused by both infectious disease and the treatment of lymphoma, among other things. In our case, the second option initially seemed more probable, since during the first hospitalisation all tests for SARS-CoV-2 were negative. The course of the disease and treatment ineffectiveness led to a reconsideration and a change of the diagnosis.

It was noted that the virus can persist for different ranges of time in the upper versus the lower respiratory system. A viral load seems to be lower in the upper than in the lower airways in patients with such a long persisting infection. Therefore, NPS tests can present a reduced sensitivity [4,5,17]. In our patient, the bronchoalveolar lavage fluid (BALF) was examined and revealed positive results for SARS CoV-2 during the third hospitalisation. Before a literature review, only NPS tests were commonly performed in SARS-CoV-2 suspected patients in our hospital.

A noteworthy question is, if a positive PCR test which persists for such a long time from the onset of symptoms reflects a viable virus or an inactive virus after a cleared infection. Cultures that were positive after incubation on proper cells prove viruses’ replication ability in patients treated with anti-CD20 medication [3,8,18]. In our hospital, performing viral cultures is not available. However, we excluded other possible causes of symptoms and radiological lesions. Moreover, cryptogenic organizing pneumonia also seems to result in inappropriate recognition because of treatment ineffectiveness and an inconsistent disease course. We believe, therefore, that ongoing SARS-CoV-2 infection with replicating virus remains a cause of described disorders and secondary organizing pneumonia.

After organizing pneumonia recognition, the typical treatment with systemic steroids was administrated. Later, the presented symptoms and the radiological pattern suggesting its exacerbation, led to intravenous pulse therapy administration followed by treatment with increased prednisone doses. Considering the insufficient effectiveness of the treatment and the relapsing positive test results for SARS-CoV-2, antiviral therapy was applied after the literature review; starting with remdesivir alone followed by the combination with casirivimab and imdevimab. There are, however, no specific recommendations for antiviral treatment in immunocompromised patients. Remdesivir has been used in COVID-19 patients, almost since the beginning of the pandemic, and is actually recommended for patients with non-severe COVID-19 at the highest risk of hospitalization [19,20]. Our patient received three courses of remdesivir because of the lack of alternative treatment options. The obvious purpose of this treatment was to decrease the viral load in his respiratory system and to try to eliminate the source of the disease. The combination of antibodies (casirivimab and imdevimab) binding to the SARS-CoV-2 spike protein was also described as effective in specific groups of patients [17]. According to the World Health Organisation (WHO) guidelines, however, casirivimab-imdevimab does not neutralize the currently circulating variants of SARS-CoV-2 and their subvariants and is not recommended [19]. As there was no possibility to investigate virus variants in our hospital, we decided to stop the therapy. After the treatment with both remdesivir in monotherapy and combined with casirivimab-imdevimab, the patient’s condition improvement was observed. In spite of that, after several weeks, fever and dyspnoea occurred again. Other medications, that were mentioned in the articles above, were unavailable in our hospital [4,8,11,12,13].

Specific anti-SARS-CoV-2 antibodies were not found in our patient despite a 3-dose vaccination and a history of recurrent infection. Only one patient described in previous literature had attained anti-SARS-CoV-2 IgM and IgG-positive serology [3]. Onishi et al. performed a study researching the impact of treatment with anti-CD20 in patients with haematological disorders on the production of anti-SARS-CoV-2 antibodies after vaccination against SARS-CoV-2. The multivariate analysis showed that the only factor significantly connected with a positive response to vaccination was a time of longer than 12 months from the last dose of anti-CD20 treatment to vaccination [21]. In our patient this time was shorter than 12 months, which could have resulted in an inadequate response to the vaccination. What is interesting, is that nonseroconventors may have lower cytokine levels and therefore a lower disease severity [5]. There are some reports, positively describing a convalescent plasma administration to anti-CD20-treated patients [6,9]. However, the evidence for this is insufficient. This procedure currently has a strong, negative recommendation [19].

During the first hospitalisation, IgA and IgG levels in our patient were within normal ranges, and IgM was mildly below normal. With the disease course, a slow decrease in IgM and IgG levels was observed. During the fourth hospitalisation, due to symptom-relapse despite antiviral treatment and the increase of prednisone doses, high-dose intravenous immunoglobulin was administrated. Taking the whole clinical picture and its complexity into consideration, it is hard to unambiguously estimate the result of such treatment, but we believe that it could have helped our patient eliminate the virus.

## 4. Conclusions

There are only a few case reports of prolonged SARS-CoV-2 infections in patients with haematological disorders after anti-CD20 treatment. This article highlights the possible clinical picture and disease dynamics. What is important is that the presence of the antigen, or even negative PCR NPS tests, for SARS-CoV-2 should not exclude this infection from the differential diagnostics. Bronchoscopy and BALF investigation should be considered. Moreover, other interstitial lung diseases, bacterial, viral, and fungal infections, and drug-related disorders should be excluded. However, their coexistence can indicate an additional complication. There is no equivocal treatment recommendation. Prolonged, exhausting symptoms do not allow, however, to leave such patients without medication. Decreasing the viral load in the respiratory system through antiviral therapy, seems to be crucial in such patients. Administration of immunoglobulin also seems to be necessary in patients with this deficiency. Such a complex approach allows the immune system to eliminate the source of the infection. In cases of recognition of organizing pneumonia, standard treatment should also be applied. More case reports are needed to assess treatment efficacy.

## Figures and Tables

**Figure 1 viruses-15-00693-f001:**
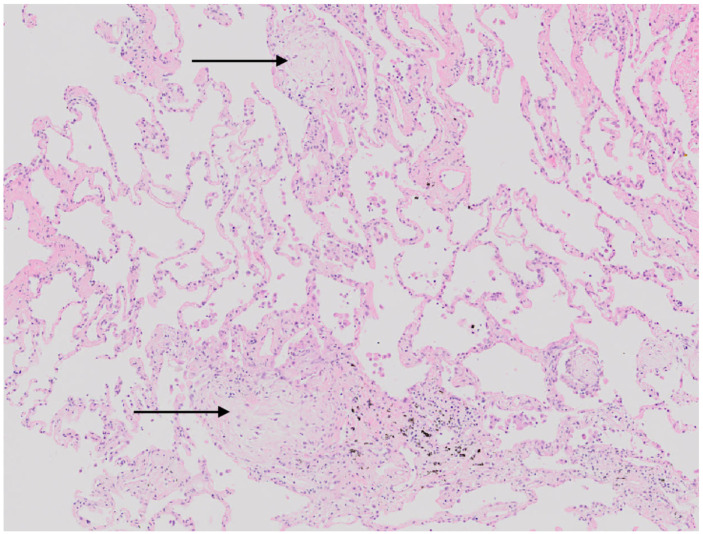
Cryobiopsy. The lung parenchyma with pale-staining fibroblast plugs that fill the airspaces—focal organizing pneumonia (black arrows). Microscopic image, H + E stain. High magnification.

**Figure 2 viruses-15-00693-f002:**
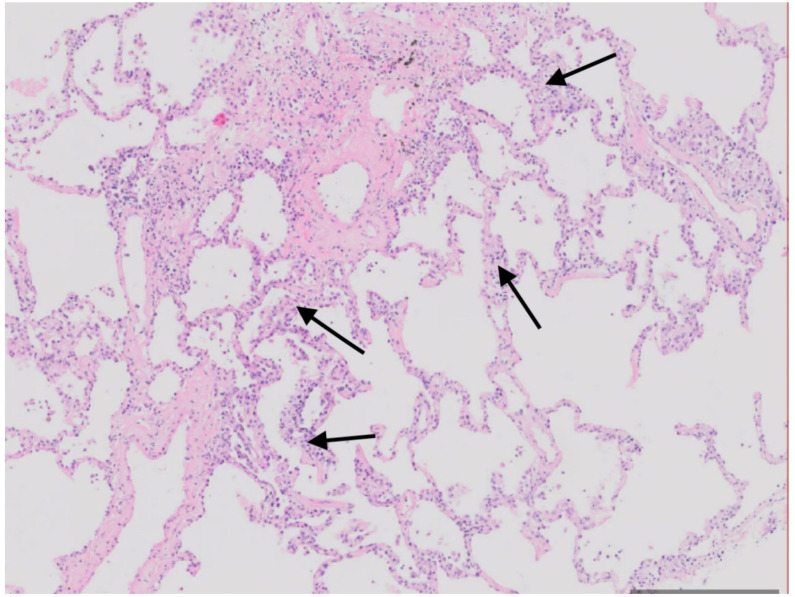
Cryobiopsy. Lung parenchyma section with a slight thickening of the alveolar septa with scattered, non-intensive, uniform inflammatory infiltration of lymphocytes (arrows) without destruction of parenchyma. Microscopic image, H + E stain. High magnification.

**Figure 3 viruses-15-00693-f003:**
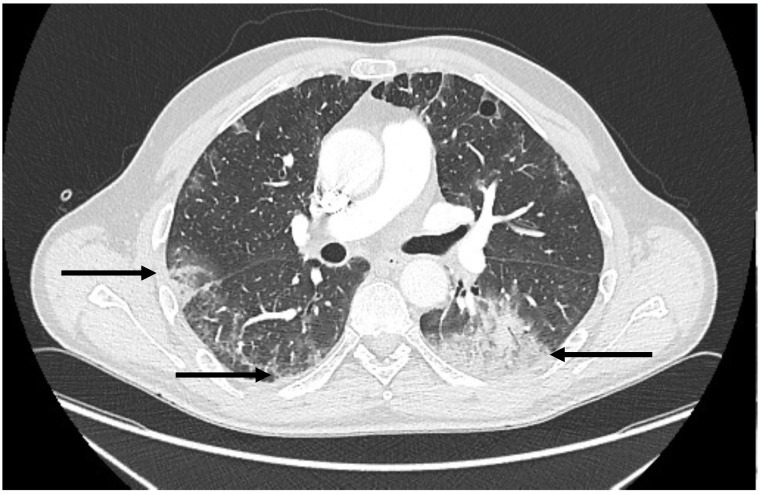
Chest CT, axial scan, lung window showing multiple bilateral and mostly peripheral areas of consolidation and ground-glass opacities (arrows) suggesting an organizing pneumonia pattern.

**Figure 4 viruses-15-00693-f004:**
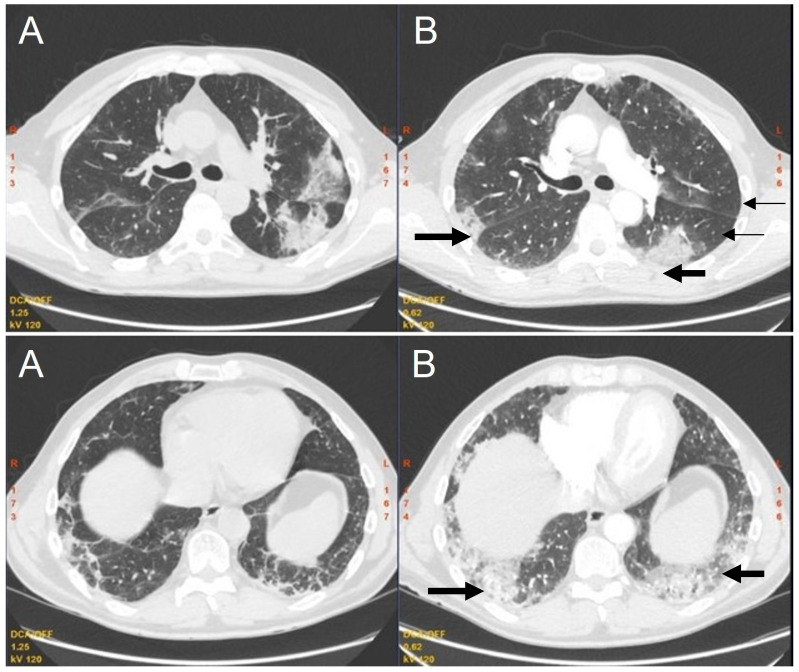
Initial chest CT (**A**) and one month follow-up chest CT (**B**) (axial scans, lung window) show partial resolution (thin arrows) of multifocal consolidation and ground-glass opacification presented in the initial chest CT-scan, and similar multifocal lung changes in new localization (thick arrows). Image variability is a typical radiological feature for organizing pneumonia.

**Table 1 viruses-15-00693-t001:** Sputum, blood, and urine microbiological examination results.

Material	Test	Result:
Sputum		
	Microbiological culture	Negative
	Mycobacterium tuberculosis complex DNA, acid-fast bacteria (AFB) smears, and cultures	Negative
	Pneumocystis jiroveci oocysts and DNA	Negative
NPS		
	The genetic material of 21 respiratory pathogens: Adenovirus: Coronavirus: HKU1, 229E, NL63, OC43; Middle East Respiratory Syndrome Coronavirus (MERS-CoV), Metapneumovirus; RSV; influenza type A subtype AH1, AH1 2009, AH3; influenza type B; parainfluenza type 1,2,3,4; Rhinovirus/Enterovirus; Bordetella pertussis; Bordetella parapertussis, Chlamydia pneumoniae, Mycoplasma pneumoniae	Negative
Blood		
	Aspergillus fumigatus galactomannan	Negative
	Aspergillus fumigatus IgG	Negative
	Cytomegalovirus (CMV) antigen pp65	Negative
	Human immunodeficiency virus (HIV) antibodies and antigen p24	Negative
	Microbiological cultures	Negative
Urine		
	Microbiological cultures	Negative
Bronchoalveolar lavage fluid		
	Microbiological culture	Negative
	Mycobacterium tuberculosis complex DNA, AFB smears, and cultures	Negative
	Pneumocystis jiroveci oocysts and DNA	Negative

**Table 2 viruses-15-00693-t002:** Pulmonary function test results, interpretation according to ERS/ATS technical standard on interpretive strategies for routine lung function tests [2].

Parameter	Result	Z-Score
FEV1%FVC [%]	81, 41	0.52
FEV1 [L; %]	3.06; 71	−2.09
FVC [L; %]	3.77; 68	−2.36
TLC [L; %]	5.98; 77	−2.58
TL, Coc Single Breath [%]	60	−3.21

**Table 3 viruses-15-00693-t003:** Summarised clinical data of all hospitalisations.

Hospitalisation	Day of Disease	Test Result	Symptoms	Suspected Diagnosis	Treatment
I	50	NPS antigen and PCR negative	Fever, cough, dyspnoea	Drug-related organizing pneumonia	Glucocorticosteroids
II	94	NPS antigen negative	Fever	Drug-related organizing pneumonia	Glucocorticosteroids’ dose increased
	103	NPS antigen positive	Fever	SARS-CoV-2 infection	5-day remdesivir therapy
III	123	NPS antigen positive	Fever, dyspnoea	SARS-Cov-2 relapse	Observation
	131	NPS antigen negative	Fever	Organizing pneumonia	Glucocorticosteroids’ dose increased
	143	NPS antigen positive	Fever	Ongoing SARS-CoV-2 infection and related to it organizing pneumonia	10-day remdesivir and 1-day casirivimab with imdevimab therapy
IV	179	NPS antigen and PCR negative	Fever	Ongoing SARS-CoV-2 infection and related to it organizing pneumonia	1-day casirivimab with imdevimab therapy and next 10-day remdesivir therapy and next intravenous immunoglobulins
	187	BALF PCR positive	Fever		

## Data Availability

Not applicable.

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
