# Peer review of "Prolonged SARS-CoV-2 Infection and Organizing Pneumonia in a Patient with Follicular Lymphoma, Treated with Obinutuzumab—Challenging Recognition and Treatment"

_viruses, 2023, doi:10.3390/v15030693_

Round 1

Reviewer 1 Report

Comments to Virus – 2242071

The manuscript describes a case report of a patient that has reached remission for follicle lymphoma after being treated with obinutuzumab. The authors describe the disease course of the patients from the day tested positive for COVID-19 and the recurrence of COVID-19 in the airways and organized pneumonia, and the use of anti-viral drugs with successful outcome.

Major comments:

The text provides a general description of the disease progression and the treatment provided for the specified patient, without stating the doses and the frequency of drug administration, which should be described in the text. Especially the doses and treatment protocol of remdesivir/casirivimab/ imdevimab – concomitant or sequential, should be provided. I believe that the authors have experience with other patients with similar conditions, and if they were treated with a similar protocol. With the same success?

The authors also need to discuss similar case reports, such as In Vivo. 2023 Jan-Feb;37(1):461-467. doi: 10.21873/invivo.13100; Respirol Case Rep. 2023 Feb 9;11(3):e01099. doi: 10.1002/rcr2.1099; Infect Drug Resist. 2023 Jan 25;16:509-519. doi: 10.2147/IDR.S396271; Infect Drug Resist. 2022 Dec 5;15:7117-7124. doi: 10.2147/IDR.S393198; and Eur J Case Rep Intern Med. 2022 Aug 31;9(8):003502. doi: 10.12890/2022_003502.

The point that I feel is lacking in the manuscript is a conclusion for the best treatment protocol of COVID-19 for immunosuppressed people, and especially those receiving anti-CD20 therapy.

The figures should all have arrows to point to the discussed observations.

Minor comments:

The title is a little bit long and should be more focused.

The abstract should provide data about the treatments given.

The last sentence in the abstract should be more conclusive.

Line 37: maybe “induce” is a better word than “induct”

Line 65: I think you meant: “the patient was presented to our hospital”

Please add space between the number and mg/mL throughout the text.

Superscript should be used instead of ^9.

Line 79 – add a Paranthesis after PCR and a point and initiate a new sentence with RT-PCR tests…

Line 89: Correct to “cryobiopsy”

Lines 182-185: The references should be within the same brackets. (and some other places)

Line 250: correct to “anti-CD20”

Author Response

1st March 2023

To the Reviewer,

we would like to thank the Reviewer for the critical review of the manuscript and for valuable comments. We have made the necessary modifications to the manuscript according to the referees’ helpful suggestions. These changes have been marked in red in the uploaded manuscript and additionally, all corrections have been provided with comments. Please find below a point-by-point response to Reviewer’s comments. We hope you find the clarifications satisfactory. Please see the attachment.

Reviewer 2 Report

The authors presented a case report of a patient with follicular lymphoma, treated with obinutuzumab, who was diagnosed with prolonged, ongoing SARS-CoV-2 infection and related to it organizing pneumonia. The recognition and the treatment were challenging which makes this case noteworthy.

The paper will be ready for publication after major revision.

The structure is good and the language is appropriate.

The abstract presents a good summary of the ideas and outcomes described in the paper.

The abstract is of reasonable length.

The authors describe the key findings of their experiments.

What is the method of obtaining samples for histopathological and H&E examination?

What are the difficulties and complication of dealing with this case?

What is the significance of (day of disease) between the clinical data of hospitalization?

In this case, is there any positive effect of SARS_covid vaccination?

What is the cause of variation in NPS and PCR test in different groups?

There is a consistent level in the references and figures.

However recent references that investigate the artificial intelligence and machine learning of in COVID-19 forecasting and diagnoses should be included such as:

A biological sub-sequences detection using integrated BA-PSO based on infection propagation mechanism: Case study COVID-19

Deep learning-based forecasting model for COVID-19 outbreak in Saudi Arabia

Artificial intelligence for forecasting the prevalence of COVID-19 pandemic: an overview

Boosting COVID-19 image classification using MobileNetV3 and aquila optimizer algorithm

Efficient artificial intelligence forecasting models for COVID-19 outbreak in Russia and Brazil

Forecasting the prevalence of COVID-19 outbreak in Egypt using nonlinear autoregressive artificial neural networks

All these references will support the introduction and discussion sections

The writing style and grammar are very good and need no improvement. 

There are no additional experiments or analyses needed for this manuscript. 

The experimental design, analytical methods and interpretation of the results are very good.

The quality of the science is and Technical English is very good.

The authors need to address the questions of practicality of this technology.

It is obvious that a great deal of effort went into this study. 

The quality of writing and science is very good.

Conclusion: What are the advantages and disadvantages of this study compared to the existing studies in this area?

The inspiration of your work must further be highlighted. Some suggested recent literatures should add.

Add future works as bullets.

Looking and wishes for the revised version.

Author Response

(The authors gave the same response as above.)

Round 2

Reviewer 1 Report

The manuscript has been improved and the authors have corrected the manuscript according to the queries.

Only a minor comment: Please Add arrows to Figure 2 and explanation in the legend.

Reviewer 2 Report

Accept.